# Examining the Effects of Agricultural Aid on Forests in Sub-Saharan Africa: A Causal Analysis Based on Remotely Sensed Data of Sierra Leone

**Qingqian He** [1,*] , **Qing Meng** [2] , **William Flatley** [3] **and Yaqian He** [3]

1 School of Political Science and Public Administration, China University of Political Science and Law, Beijing 102249, China
2 Department of International Relations, Tsinghua University, Beijing 100084, China; q-meng16@mails.tsinghua.edu.cn
3 Department of Geography, University of Central Arkansas, Conway, AR 72035, USA; wflatley@uca.edu (W.F.); yhe@uca.edu (Y.H.)
* Correspondence: qingqian@cupl.edu.cn; Tel.: +86-137-187-467-81

**Abstract:** In sub-Saharan Africa, extreme poverty highlights the tension between development aid and the environment. Foreign aid is considered one of the most important factors affecting forest health in this region. Although many studies have empirically examined the effects of different kinds of foreign aid on forests, few have investigated the potential impact of agricultural aid. This study investigated the causal effects of agricultural aid on forests in Sierra Leone, a country that relies heavily on agricultural products. We constructed a fine-grained (16-day) indicator of forest growth from 2001 to 2015 by combining remotely sensed data of land cover and the Normalized Difference Vegetation Index. The high frequency of forest growth data enables us to trace the dynamic causal process. To reduce the confounding effects of heterogeneity, we applied a difference-in-difference design with data at the sub-national level to estimate the causal effect. This study provides robust empirical evidence that foreign agricultural aid harms forests both in the short term (i.e., 16 days) and long term (i.e., years) in Sierra Leone. Agricultural aid projects with agricultural development as their primary objective or aid projects without specific objectives lead to the highest levels of forest degradation.

**Keywords:** agricultural aid; forest protection; remote sensing; environmental influence





## 1. Introduction

The reconciliation of environmental conservation and human development is challenging and difficult to achieve [1–4]. In 1992, during the Rio Earth Summit, developed nations, such as the United States, the members of the European Community, Canada, and Japan, announced that they would increase their financial assistance to developing countries for the protection of the environment [5]. Over the last thirty years, many aid agencies have made efforts to incorporate conservation objectives into their development aid projects, promising to benefit both the environment and people [6–8]. In 2015, the UN Sustainable Development Summit set up more than 17 Sustainable Development Goals (SDGs) and addressed the necessity of reducing poverty, advancing social equity, and simultaneously ensuring environmental protection. Thereafter, developing foreign aid that is more environmentally friendly and avoids disrupting biodiversity in developing countries has attracted increasing attention from scholars and policymakers in the field of the global governance of aid [8–11].

Forests cover about a third of the Earth's land area and are essential to the health of the global environment [12]. Forest degradation is a reduction in tree density and/or an increase in disturbance in a forest that can result in the loss of forest products and

forest-derived ecological services [13]. It not only contributes to greenhouse gas emissions and thus accelerates the pace of climate change, but it is also directly associated with the decline of biodiversity [14–19]. The existing literature identifies a wide range of proximate or direct drivers and underlying or indirect causes of forest degradation, such as selective felling, fuelwood collection, shifting cultivation, road construction, and wildfires [20–23]. Foreign aid is a concern because it is an important resource for developing countries to accelerate the process of economic development. Although many studies have empirically examined the effects of different kinds of foreign aids on forests, such as development aid [8,24], infrastructural aid [25], conservation aid [9,11], forestry aid [10], and bilateral aid [10,26], fewer researchers have investigated the potential impact of agricultural aid.

More than 75% of the poor people in the developing world live in rural areas, and most of them depend directly or indirectly on agriculture for their livelihood [27,28]. Numerous studies have demonstrated that the development of agriculture is particularly effective in reducing poverty [29,30]. However, a number of developing countries, especially countries in sub-Saharan Africa, lack enough resources to provide the required investment to their agricultural sectors and farmers [31,32]. As such, foreign agricultural aid is important for developing countries to improve their agricultural growth and productivity. Since the early 1970s, international donors have donated millions of dollars (USD) to farmers and agricultural sectors in poor regions around the world [33]. However, in sub-Saharan Africa, where much of agricultural aid is targeted, there is great pressure on forests due to cutting for firewood, charcoal and logging, and the expansion of agriculture [19,20,34]. While the effects of agricultural aid on agricultural development and poverty reduction have been documented [35–37], the evidence of its impact on the environment is relatively sparse.

From a theoretical perspective, the predictions with respect to the effects of agricultural aid on forests in sub-Saharan Africa are mixed. One set of hypotheses highlight that agricultural aid has already incorporated environmental concerns [33], which could encourage forest protection. Agricultural expansion is mainly caused by demand for more cropland for food, fiber, and biofuel production at the expense of other land cover types, such as forests [38], resulting in the diminishing of the provision of ecosystem goods and services. However, since the mid-1990s, the aim of agricultural aid is no longer merely improving total agricultural growth and output, but focusing more on increasing the growing capacity of the agricultural land (i.e., intensification). The New Partnership for Africa's Development (NEPAD) clearly pointed out that sustainable agriculture could be achieved by using a variety of techniques, such as crop rotation, soil enrichment, and natural pest predators [39]. Agricultural aid aiming to improve soils, offer improved seeds, provide more efficient irrigation (e.g., easier accessibility to water [33,35]), and supply farmers with appropriate new technologies (e.g., agricultural machinery and equipment [33,35]) can increase the productivity of a given size of farmland. For instance, studies have found that specific types of irrigation (e.g., flood irrigation) positively increase soil water and soil nutrient concentrations in the upper soil layer [40,41]. These can increase croplands intensity rather than expansion, discourage the cultivation of larger acreage, and thus benefit other vegetation types, including forests. In sum, agricultural aid may benefit forests.

Alternatively, agricultural aid to sub-Saharan countries may harm forests. Agricultural aid for the development of water resources is mainly directed towards large-scale projects on irrigation, reservoirs, hydraulic structures, and groundwater exploitation [33]. However, these projects can directly contribute to damaging forests. For example, hydroelectric dams can change a river's hydrological cycle, which in turn affects land cover and vegetation outcomes, including forests, resulting in a change of the ecology of the floodplain and the spatial distribution of flora and fauna [42,43]. Diverting water to a downstream area, irrigation dams can intensify the conversion of forested land into land that is suitable for agriculture production [44]. In addition, since fertilizer and pesticides are strategically important in increasing agricultural productivity and ending hunger [35], international donors continuously provide them for recipient countries [33,35]. Unfortunately, the fertilizers and pesticides used in agriculture have negative consequences for the environment.

Additionally, they are not limited to agricultural systems since the residues of fertilizers and pesticides can be dispersed through the air, leach into the soil, groundwater, and run-off into surface water [45]. Various studies have found that the excessive use of fertilizers and pesticides has led to biodiversity loss and ecosystem degradation [46,47]. Moreover, when discussing the relationship between agricultural aid and forests, one cannot ignore the reality of poverty in most sub-Saharan countries. Smallholders are the main driving force of African agricultural expansion and forest losses [48,49]. Without sufficient off-farm jobs in urban areas, people may continue to exploit forests. Harvesting for timber, firewood, charcoal, grazing and even converting forests to agricultural land (e.g., cropland) to enlarge planting area for crops (e.g., rice) are the major drivers of deforestation and forest degradation [20,21]. These human activities may be accelerated with the continued financial support from aid for agricultural development. Finally, governance quality has been found to be critical in protecting forests [50–52]. However, the quality of governance in most sub-Saharan countries is quite poor. Agricultural aid is mainly delivered by the central or local governments of the recipient countries. If their overall governance capacities are low, aid to those countries may be easily abused or misappropriated to conduct more commercially valuable activities compared to agriculture, such as logging. All in all, agricultural aid may harm forests.

As discussed above, the impacts of agricultural aid on forests in the sub-Saharan region are quite ambiguous. It is unclear whether there is an intrinsic tension between agricultural aid and forest protection. Furthermore, outcomes may depend heavily on how agricultural aid is designed and implemented. We address this knowledge gap by empirically analyzing the effects of foreign agricultural aid on forest growth using Sierra Leone as a case study. Located in West Africa, Sierra Leone is one of the poorest countries in the world and has long suffered from brutal poverty [53]. It is ranked 181 out of 189 countries in the 2019 Human Development Index Ranking. Although agriculture is the largest sector of the economy and employs more than 80% of the population, which is about 7,092,113 as of the 2015 census [54], nearly half the population is food insecure and half of all child deaths are attributable to malnutrition. This was once attributed partly to Sierra Leone's poor soil. Ferralsols, oxisols, inceptisols, entisols, spodosols, and ultisols are the primary soil types in Sierra Leone [55,56]. The international soil reference and information center (ISRIC) shows that soil organic carbon (SOC) contents in most regions of Sierra Leone are between 30–60 tons per hectare, with some regions in the northern and western parts of the country containing between 75–105 tons per hectare [57]. The country is highly dependent on foreign aid, including agricultural aid, especially after a ten-year civil war. Sierra Leone lost about 1,710,000 hectares of tree cover between 2001 to 2020, equivalent to a 30% decrease in tree cover and 808 million tons of $CO_2$ emissions [58], which is alarming for its ecosystems and biodiversity.

In order to explore the role of agricultural aid in forests in Sierra Leone, we first developed a fine-grained indicator of forest growth at the chiefdom level, the lowest administrative unit, measured at a 16-day frequency using satellite-based data of land cover and the Normalized Difference Vegetation Index (NDVI) from 2001 to 2015. With this dataset, we then applied the difference-in-difference (DID) design to estimate the causal effect of agricultural aid on forests in order to rule out the influence caused by the potential confounders, such as precipitation, temperature, governance quality, and poverty level. To our knowledge, the study is the first attempt to explore the environmental consequences of agricultural aid. The characteristics of high-frequency remotely sensed data with geocoded agricultural aid data not only make it possible to conduct a sub-national analysis, leading to more rigorous findings, but also enable us to observe the dynamic influence of agricultural aid on forests across time. There are three specific questions addressed in the study: (1) Does agricultural aid benefit or harm forest growth? (2) Does the relationship between agricultural aid and forest growth change over time? (3) Do different kinds of agricultural aid have different effects on forest growth? These questions are significant for forest protection since they could tell us whether or not additional interventions are

needed during the implementation of agricultural aid, and if they are, when and how. In addition, pioneered by Alexander Mather, research has applied forest transition models to explain the evolution of forest land cover in developing countries, such as Vietnam [59], Latin America [60], India [61], and China [61,62]. Lambin (2009) once summarized two fundamental forces in general, i.e., socio-ecological feedbacks and socio-economic factors, in understanding the reasons behind forest decline and recovery [59]. This study contributes to this broad topic by focusing on the influence of an exogenous socio-economic factor, i.e., foreign agricultural aid, on forests in the sub-Saharan region.

## 2. Materials and Methods

### 2.1. Materials

2.1.1. Agricultural Aid Data

While most existing studies mainly focus on national level aid inflows, we concentrated on sub-national level aid (chiefdom level) in this study. The total amount of aid received by a country reveals nothing about how aid is actually allocated spatially, and how much aid is provided to each part of the country. Aid, including agricultural aid, is eventually allocated at local scales and generates local outcomes. Macro-scale findings may be misleading when the actual distribution of aid is neglected. With more precise geographical information on whether agricultural aid is present or absent in a specific area combined with the measures of forest growth conditions, we can better investigate the possible causal relationship between agricultural aid and forests.

Chiefdoms are the lowest formal administrative units in Sierra Leone [53,63]. Since the land and other important resources, such as diamonds, are mainly regulated by the authorities (i.e., chiefs) in chiefdoms [53,63], it is suitable to examine the impact of agricultural aid on forests at the chiefdom level. Additionally, as other aid-recipient countries, Sierra Leone receives agricultural aid projects at multiple administrative levels, from the national level to the provincial and the district levels, and then further down to the lowest level of chiefdoms. The hierarchical system of aid allocation denies the assumption that agricultural aid is equally distributed. It in turn may generate false causal evidence when agricultural aid is allocated in one area and forest degradation occurs in another area in the same province or district. This provides further evidence that the best strategy for the study is to focus on the lowest level of administration, i.e., chiefdoms.

We obtained the geo-coded agricultural aid data of Sierra Leone between 2001 and 2015 from the AidData project, which is managed by a research lab at the College of William & Mary's Global Research Institute in the United States [64]. Founded in 2009, the AidData has provided access to aid activity records from more than 90 donors (e.g., the United States and EU member countries) and multilateral organizations (e.g., World Bank and OECD) from 1945 to the present. The AidData records usually cover the name of donors and recipients, name of aid projects, types of aid projects, purpose of aid projects, amount of aid projects, start time of aid projects, end time of aid projects, among other records. Cooperating with Uppsala University in Sweden, AidData also tags aid activities with geographic coordinates to pinpoint aid projects to geographic locations [65]. This is the major advantage of the AidData project compared to other official sources of aid statistics, such as the Creditor Reporting System (CRS), which is the central database for foreign aid compiled by OECD's Development Assistance Committee (DAC). The AidData database is publicly available to policymakers, practitioners, and academic researchers to make development and foreign aid more transparent, accountable, and effective.

In this study, we investigated the impact of receiving agricultural aid projects on forest growth. Thus, the treatment variable was dichotomous. In total, there were 149 chiefdoms in Sierra Leone before August 2017. Among them, 58 received chiefdom-level agricultural aid projects (Figure 1a). The number of agricultural aid projects received ranged from 1 to 5, and the majority of chiefdoms received 1 project (Figure 1a). The starting time of these aid projects was concentrated around 31 December 2007, and the earliest one was 2 October 2006. The time period of implementation of these aid projects ranged from

1 year to 7.7 years, but most of them were around 6.5 years (Figure 1b). The agricultural aid type "Rural Finance and Community Improvement Project (RFCIP)" occupied the largest proportion of agricultural aid to chiefdoms (Table 1). "Promoting Agriculture, Governance and the Environment (PAGE)" was second, followed by "Sustainable Nutrition and Agricultural Promotion (SNAP)" (Table 1). There is an obvious variation in the types of agricultural aid projects received by each chiefdom (Figure 1c). This information makes it possible to examine the overall impact of agricultural aid, the influence of agricultural aid across time, and the effects of different kinds of agricultural aid on forests.

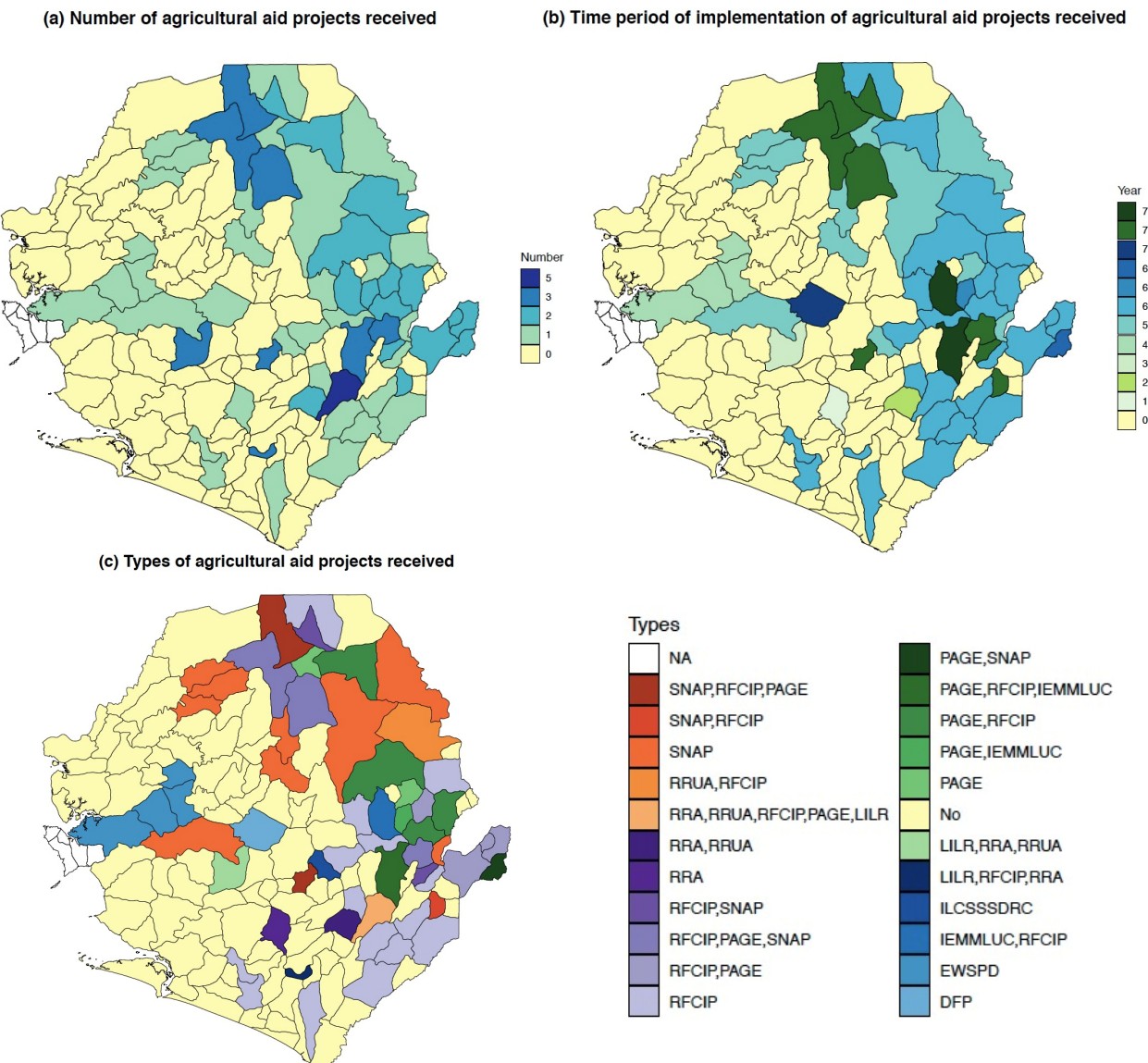

**Figure 1.** Description of Sierra Leone's chiefdom-level agricultural aid projects from 2001 to 2015. (**a**) Number of agricultural aid projects received; (**b**) Time period of implementation of agricultural aid projects received; (**c**) Types of agricultural projects received (locations in white are Western Area Urban and Western Area Rural, which are not chiefdoms. ILCSSSDRC: Improving Living Conditions and Strengthening of Social Stability and Decentralization in Rural Communities of Former Civil War Areas in Southern Sierra Leone; PAGE: Promoting Agriculture, Governance and the Environment; RRA: Refugees in Rural Areas in Sierra Leone; RFCIP: Rural Finance and Community Improvement Project; SNAP: Sustainable Nutrition and Agricultural Promotion; EWSPD: Empowering Women for Sustainable Peace and Development; DFP: Diversified Food Production; No: do not receive any types of agricultural aid.)

**Table 1.** Types of agricultural aid projects received in chiefdoms in Sierra Leone from 2001 to 2015.

| Group of Agricultural Aid | Type of Agricultural Aid | Number of Projects |
|---|---|---|
| Agricultural aid without specific objectives | Sustainable Nutrition and Agricultural Promotion, SNAP | 18 |
| Agricultural aid with agricultural development as major objectives | Diversified Food Production, DFP | 1 |
| | Rural Finance and Community Improvement Project, RFCIP | 36 |
| Agricultural aid with environmental concerns | Improving Environmental Management and Mitigating Land-Use Conflicts in alluvial Diamond Field in Sierra Leone, IEMM | 3 |
| | Promoting Agriculture, Governance and the Environment, PAGE | 19 |
| Agricultural aid with local governance concerns | Improving Living Conditions and Strengthening of Social Stability and Decentralization in Rural Communities of Former Civil War Areas in Southern Sierra Leone, ILCSSSDRC | 1 |
| Agricultural aid concerning specific social groups | Local Integration of Liberian Refugees, LILR | 3 |
| | Empowering Women for Sustainable Peace and Development, EWSPD | 4 |
| | Refugees in Rural and Urban Areas in Sierra Leone, RRUA | 4 |
| | Refugees in Rural Areas in Sierra Leone, RRA | 5 |
| | Total | 94 |

Data Source: AidData project. Types of agricultural aid projects are based on the names and purposes of agricultural aid projects categorized by AidData.

### 2.1.2. Forest Cover and Vegetation Index Data

To identify forest cover in Sierra Leone, we used the Moderate-resolution Imaging Spectroradiometer (MODIS) land cover dataset (MCD12Q1) from 2001 to 2015. MODIS MCD12Q1 has been widely used to detect land cover and land use change, including forest dynamics [66,67]. The MODIS MCD12Q1 comprises annual maps of land cover and land use with a spatial resolution of 500 m [68]. The classification system of MCD12Q1 used in this study was the International Geosphere-biosphere Program (IGBP) classification system, which includes 17 classes: water bodies, evergreen needleleaf forests, evergreen broadleaf forests, deciduous needleleaf forests, deciduous broadleaf forests, mixed forests, closed shrublands, open shrublands, woody savannas, savannas, grasslands, permanent wetlands, croplands, urban and built-up, cropland and natural vegetation mosaic, snow and ice, and barren or sparsely vegetated. Following previous studies [69], we aggregated all forest types, including evergreen needleleaf forests, evergreen broadleaf forests, deciduous needleleaf forests, deciduous broadleaf forests, and mixed forests, as one forest type and excluded non-forest classes.

To measure forest growth conditions, we used the Normalized Difference Vegetation Index (NDVI) data from the MODIS Vegetation Indices product (MOD13A1) from 2001 to 2015. NDVI was calculated from the visible red and near-infrared light reflected by Earth's surface [70]. The range of NDVI is from −1 to +1. Water is generally associated with negative NDVI values, and bare soil with values near zero (0.1 or less). Sparse vegetation, such as grasslands, may result in moderate NDVI values (~0.2 to 0.5), while dense vegetation, such as forests, show high NDVI values (~0.6 to 0.9) [71,72]. A number of studies have used NDVI to monitor forest growth conditions [73–77]. Yet, NDVI values tends to saturate in dense canopies, such as temperate and tropical forests [78]. Fortunately, it did not show this saturation in our study area (Figure A1). The 16-day MOD13A1 product had a spatial resolution of 500 m, which is consistent with MODIS land cover data. Based on the quality layer along with the NDVI data, we excluded the MODIS NDVI values with a low quality that may be contaminated by cloud cover.

### 2.2. Methods

### 2.2.1. MODIS Data Pre-Processing

Based on MODIS land cover data, we retained the forest areas and excluded the non-forest areas in Sierra Leone each year from 2001 to 2015. We then extracted the NDVI values for each grid cell only within the forest areas, and sum the NDVI values over all forest grid cells ($sum_{forest\ NDVI}$, hereafter) in a given chiefdom based on the administrative boundaries.

We chose the summation of NDVI values rather than mean values to take account of the cases where forests in an entire grid cell are degraded to/restored from other land cover types. As MODIS NDVI has a temporal resolution of 16 days, we obtained 23 $sum_{forest\ NDVI}$ values for each chiefdom each year. As such, we obtained 345 $sum_{forest\ NDVI}$ values from 2001 to 2015. Forest degradation (restoration) occurring in a chiefdom was identified by the decrease (increase) in the $sum_{forst\ NDVI}$ value at the same time across years in that chiefdom. As there was no forest cover shown in MODIS land cover data for 14 chiefdoms during 2001 and 2015, we obtained $sum_{forst\ NDVI}$ values for 135 out of 149 chiefdoms. Figure 2 is an example that shows the summation of 23 values of $sum_{forest\ NDVI}$ over a year for each of 135 chiefdoms in Sierra Leone. Our analyses focused on these 135 chiefdoms.

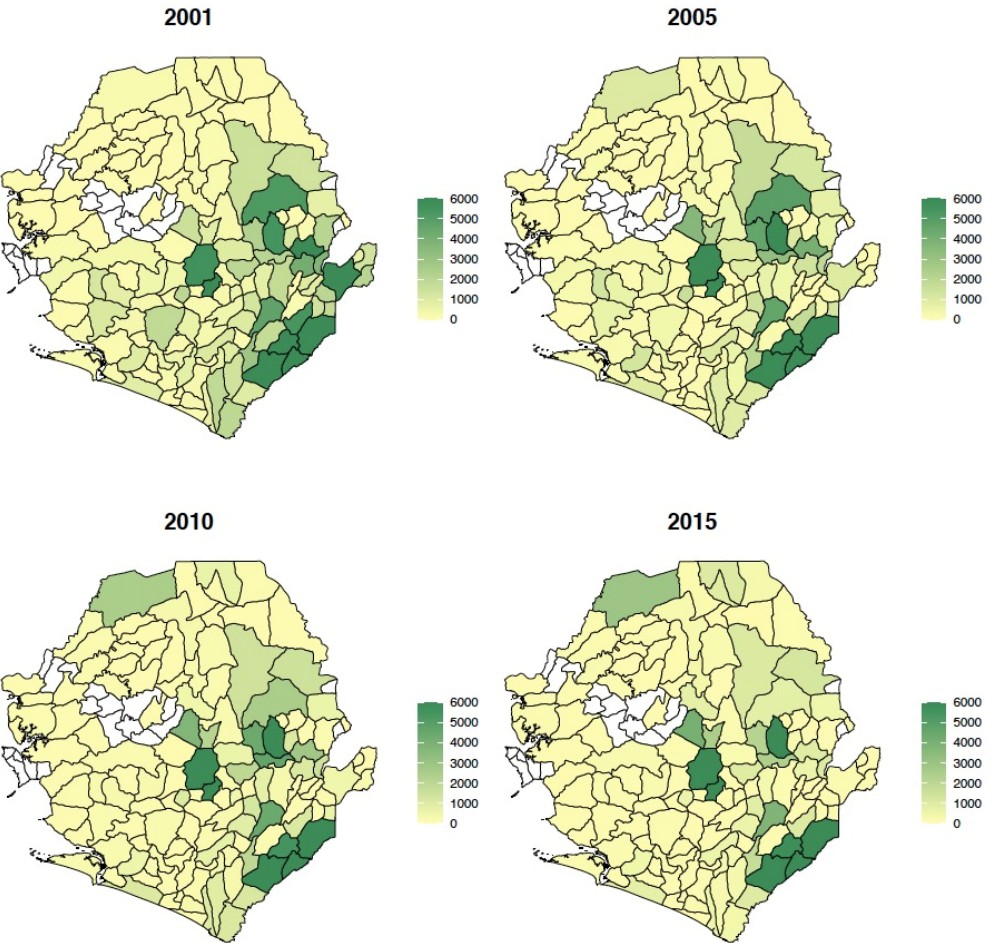

**Figure 2.** The summation of 23 values of $sum_{forest\ NDVI}$ over a year for each chiefdom in Sierra Leone in 2001, 2005, 2010, and 2015 (Locations in white are Western Area Urban and Western Area Rural that are not chiefdoms, and are 14 chiefdoms without forest cover shown in MODIS land cover data during 2001 and 2015).

### 2.2.2. Difference-in-Difference Approach

To investigate the impact of agricultural aid on forests, the study used a difference-in-difference (DID) approach, which is a quasi-experimental design that makes use of longitudinal data from treatment and control groups to obtain an appropriate counterfactual to estimate a causal effect [79–83]. The treatment group receives a specific intervention, while the control group does not. The effect of the intervention is estimated by comparing the changes in outcomes over time between the treatment and control groups (Figure 3). The advantage of the DID approach is that it applies the logic of natural experiment, thus makes it possible to avoid the interference of omitted or unobserved variables and prevent

endogeneity problems, such as reverse causation caused by a selection bias. In this study, receiving or not receiving agricultural aid projects was a quasi-experimental intervention.

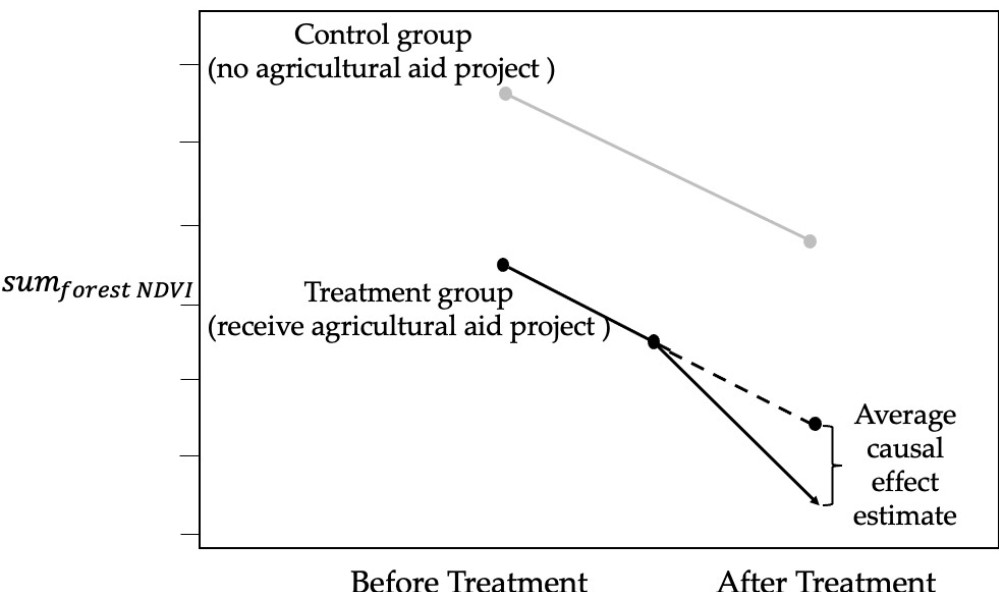

**Figure 3.** Illustration of the difference-in-difference approach.

Unlike random experimental designs, which rely on an exchangeability between the treatment and control groups, DID is based on a less strict exchangeability assumption, i.e., in the absence of intervention, and the differences between the treatment and control groups are similar over time. That is to say, in a DID design, the treatment and control groups are not randomly assigned so that the group-specific means might differ in the absence of treatment. However, as long as the two groups are changing in the same way over time, or in other words, the differences between two groups are constant across time, it can be distinguished by deducting group-specific means of the outcome of interest. Thus, the remaining difference between these group-specific differences can reflect the causal effect of interest [83]. Because of this less strict requirement on the treatment and control groups, DID designs have been widely applied in social science to estimate the impact of institutions and policies [79–83]. To satisfy the parallel time trend requirement, many studies use the geographically closest approach to match treatment groups with control groups [84,85]. The idea behind this strategy is simple, following Tobler's first law of geography [86,87]. It states that everything is related to everything else, but near things are more related than distant things [86]. Therefore, this study took chiefdoms receiving agricultural aid projects as the treatment group, and the geographically closest chiefdoms not receiving agricultural aid projects in the same district were the control groups. The reasons include the followings: Firstly, forests are strongly affected by regional variation in climate, landforms, and soils. These variations in natural conditions in geographically adjacent chiefdoms are likely to remain relatively constant over the time span we are examining. Neighboring chiefdoms may also experience similar economic shocks and other human activities that affect forests. Thus, geographic adjacency is important for the parallel time trend assumption to hold. Secondly, chiefdoms in different districts are more likely to experience different time trends. It may be due to the possibility of the district-level changes of politics and policies, variation of the district- and upper-level agricultural aid reception, and other endogenous or exogenous district-level shocks. Hence, choosing the control chiefdom for each treated one from the same district was a better strategy.

### 2.2.3. Models

Because $sum_{forest\ NDVI}$ is time series data repeated every 16 days, we calculated the mean $sum_{forst\ NDVI}$ value of the same time period (*n*th 16-day) across five years before the treated years as the before outcome. The five-year window for the before outcome is chosen to consider the stability of forest growth conditions before treatment. The after outcome is the $sum_{forst\ NDVI}$ of the same time period (*n*th 16-day) across the treated years. As mentioned previously, for those chiefdoms that receive chiefdom-level agricultural aid projects, many received multiple projects, raising the problem of repeated treatments. However, in all the cases that multiple chiefdom-level projects are received by the same chiefdom, these projects have overlapping treatment periods. This means that the start time of some agricultural aid projects overlap with the end time of the other agricultural aid projects in those chiefdoms, or vice versa. This makes it difficult to investigate the impact of the amount of agricultural aid to different chiefdoms on forest cover. However, since the purpose of the paper was to examine whether the presence or absence of agricultural aid projects makes a difference in forest cover, we counted the multiple projects as one treatment lasting for the combined duration. The following is the mathematical expression of the difference-in-difference approach for this study:

$$\delta_{it} = \Delta Y^T_{i,t+p} - \Delta Y^C_{j,t+p} \tag{1}$$

$$= \left( Y^T_{i,t+p} - Y^T_{i,t} \right) - \left( Y^C_{j,t+p} - Y^C_{j,t} \right) \tag{2}$$

where chiefdom *i* is the chiefdom that received chiefdom-level agricultural aid project at the *p*th time period after time *t*, while chiefdom *j* is the chiefdom that did not receive chiefdom-level agricultural aid project before and after time *t*. $Y^T_{i,t+p}$ is the outcome of $sum_{forst\ NDVI}$ in chiefdom *i* at $t + p$, the *p*th period since it has received agricultural aid. $Y^T_{i,t}$ is the outcome before aid (i.e., the average $sum_{forst\ NDVI}$ over the same time in the previous five years before treatment). $Y^C_{j,t+p}$ is the $sum_{forst\ NDVI}$ of chiefdom *j* at time $t + p$, and $Y^C_{j,t}$ is the average $sum_{forst\ NDVI}$ of the same time in the previous five years before chiefdom *i*'s treatment. Our data allowed us to estimate $\delta_{i,t+p}$ for *p* = 1st, 2nd, 3rd, 4th, . . . 345th 16-day.

We specified the following regression models of the difference-in-difference approach to estimate the causal effect of agricultural aid on forests:

$$\textbf{Model } 1\ Y_{it} = \beta_0 + \beta_1 D_i + \beta_2 T_t + \gamma (D_i * T_t) + \alpha_i + \eta_t + \varepsilon_{it} \tag{3}$$

$$\alpha \sim N(0, {\sigma_\alpha}^2) \tag{4}$$

$$\eta \sim N(0, {\sigma_\eta}^2) \tag{5}$$

$$\varepsilon_{it} \sim N\left(0, \sigma_\varepsilon^2\right) \tag{6}$$

where *i* refers to chiefdom *i*, and *t* is the indicator of time (in this study, the frequency is 16 days). $D_i$ is an indicator of whether chiefdom *i* is treated, and $D_i = 1$ if the chiefdom is treated (i.e., receiving agricultural aid project); otherwise, it is 0. $T_t$ is also a binary variable with $T_t = 1$ if time *t* is after the treatment time and $T_t = 0$ otherwise. The parameter of $\gamma$ is the quantity of interest as the DID estimator. $\alpha_i$ and $\eta_t$ control unobserved chiefdom-specific and time-specific characteristics that may affect the assignment of agricultural aid in different chiefdoms and time periods, and also be related to forest growth. We regard Model 1 based on Equation (3) with a more general hierarchical model (or mixed-effect model) and apply a restricted maximum likelihood (REML) approach to estimate the parameters using the R package lme4.

To trace the time-varying causal effect of agricultural aid on forests, we specified a varying-coefficient specification as follows:

$$\textbf{Model } 2\ Y_{it} = \beta_0 + \beta_{1t} D_i + \beta_{2t} T_t + \gamma_t (D_i * T_t) + \alpha_i + \eta_t + \varepsilon_{it} \tag{7}$$

$$\beta_{1t} \sim \text{N}\left(\beta_1,\ \sigma_\beta^2\right) \tag{8}$$

$$\beta_{2t} \sim \text{N}\left(\beta_2,\ \sigma_\beta^2\right) \tag{9}$$

$$\gamma_t \sim \text{N}\left(\beta_1,\ \sigma_\gamma^2\right) \tag{10}$$

$$\alpha \sim \text{N}\left(0,\ \sigma_\alpha^2\right) \tag{11}$$

$$\eta \sim \text{N}\left(0,\ \sigma_\eta^2\right) \tag{12}$$

$$\varepsilon_{it} \sim \text{N}\left(0,\ \sigma_\varepsilon^2\right) \tag{13}$$

where $\gamma_t$ is the DID estimators of the causal effects of agricultural aid on forests across time. Model 2 is similar to Model 1, except for $\beta_1$, $\beta_2$, and $\gamma$ varying with time $T$. We also use REML to estimate the model. As climate is highly relevant to forest growth, we also added addition control variables of temperature and precipitation data in Model 1 and Model 2. The results are consistent with our original conclusions (Tables A1 and A2, and Figure A2 in Appendix A).

## 3. Results and Discussion

### 3.1. Overall Impact of Agricultural Aid on Forests

We estimated the general effect of agricultural aid on forests with the whole sample. The remote sensing technique provides the forest growth conditions at a 16-day frequency. We grouped data according to chiefdom (i.e., whether a chiefdom receives treatment or not) and time (i.e., before or after treatment). By doing so, we created a dataset with a 16-day frequency that contained more than 30,000 observations in total.

Table 2 shows the DID estimates and their uncertainties with control variables, i.e., unobserved chiefdom-specific and time-specific characteristics that may affect the impact of agricultural aid. Model 1 demonstrates that the causal effect of agricultural aid on $sum_{forest\ NDVI}$ is negative and highly significant. The negative impact is around $-0.071$, which indicates that, if an agricultural aid project in a specific chiefdom increases one unit, $sum_{forest\ NDVI}$ will decrease around 0.071 unit. This shows that the effect of agricultural aid on forests is quite obvious, and agricultural aid to chiefdoms is responsible for forest degradation. All of these imply that strengthening the linkages between poverty reduction and the management of forest conservation is a key challenge to reducing poverty in Sierra Leone.

**Table 2.** Causal effect of agricultural aid on forests.

|  | **Model 1** | |
| --- | --- | --- |
|  | **Estimate** | **Std. Error** |
| **Fixed effects** | | |
| D (receive aid) | 0.159 | 0.269 |
| T (Time) | 0.032 *** | 0.004 |
| D × T (receive aid × Time) | −0.071 *** | 0.006 |
| Intercept | −0.205 | 0.239 |
| **Random effects** | | |
|  | **Variance** | **Std. Dev** |
| Chiefdom | 0.856 | 0.925 |
| Time | 0.007 | 0.085 |
| Residual | 0.077 | 0.278 |
| **Number of Observation** | 31,164 | |

Note: *** $p < 0.01$.

### 3.2. Dynamic Impact of Agricultural Aid on Forests

We further explored the impact of agricultural aid on forests to answer whether there is a shift in the relationship between agricultural aid and forest growth. We conducted a sub-sample analysis with 16-day frequency after the agricultural aid was introduced. Since the longest chiefdom-level agricultural aid lasts around 7.7 years, we examined the impact of agricultural aid on forests from the first 16-day period (i.e., 0.5 months) to 178th 16-day period (i.e., 84 month).

Figure 4 presents the results from the difference-in-difference analyses based on Model 2 for each 16-day time point. We found that there is no shift in the relationship between agricultural aid and forest growth. Agricultural aid negatively and significantly influences forest growth in both the short term (i.e., 16-day frequency) and long term (i.e., years). After about 70 months, the error bounds (i.e., the shaded green areas) are slightly larger than earlier time periods. This may be due to the fact that the number of observations is not large enough to reveal reliable empirical evidence about the causal effect since not many agricultural aid projects last this long. In sum, our results show that the impact of agricultural aid on forest growth is negative and highly significant across time, and a turning point does not appear at least during the period of implementation of agricultural aid from 2001 to 2015.

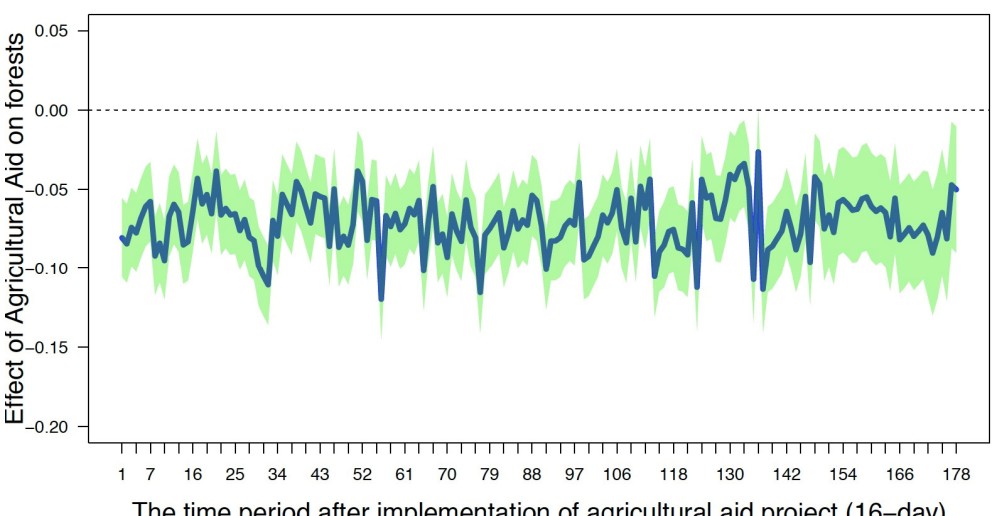

**Figure 4.** Causal effects of agricultural aid on $sum_{forest\ NDVI}$ across time (the shaded green areas represent the 95% confidence interval).

### 3.3. Impacts of Different Types of Agricultural Aid on Forests

The goals of agricultural aid vary greatly. They range from food security and agricultural production to rural finance support and governance policy on administration and environmental protection. Since the purposes of different types of agricultural aid are different, their impacts on forests may be also diverse. Do agricultural aid projects with more social and environmental concerns damage forests less than those that have agricultural development as their sole purpose? In this section, we investigate the effects of different types of agricultural aid on forests. We distinguish types of agricultural aid based on the names and purposes of agricultural aid projects categorized by AidData. In the matched data, there were 30 chiefdoms that receive only one type of agricultural aid. More specifically, 14 chiefdoms receive RFCIP, 9 chiefdoms receive SNAP, 2 chiefdoms receive PAGE, 1 chiefdom receives RRA, 1 chiefdom receives ILCSSSDRC, 2 chiefdoms receive EWSPD, and 1 chiefdom receives DFP. With this information, we could explore the influence of seven different types of agricultural aid on forest growth.

Table 3 presents the results of our data analysis of the different types of agricultural aid from Model 1. Agricultural aid named ILCSSSDRC positively influences forest growth conditions. The impact is around 0.031, which means that with an increase of one unit of

ILCSSSDRC, the $sum_{forest\ NDVI}$ will increase by 0.031 units, indicating an improvement in forest growth conditions. ILCSSSDRC's purpose is to improve the quality of governance in rural communities. The results therefore suggest that agricultural aid can benefit forests through supporting rural governance. This is consistent with the findings of many other studies on governance quality and forest protection [46–48]. The most negative impact on forests comes from the agricultural aid of DFP and SNAP. DFP aims at promoting diversified food production. Additionally, SNAP mainly focuses on some unspecified objectives on agricultural promotion, such as education and basic nutrition. RFCIP, which provides specific supports for agricultural development of rural communities, has a moderate negative impact on forest growth. Its impact is about −0.035. PAGE, EWSPD, and RRA have the smallest negative impacts on forest growth conditions. PAGE aims at promoting agricultural development with a view on governance and the environment. RRA has a rather specific purpose to provide clean water and sanitation to refugees. Additionally, EWSPD's purpose is to empower women in rural areas in Sierra Leone. As a whole, agricultural aid with social and environmental concerns degrades forests less than those that have agricultural development as the major goal.

**Table 3.** Causal effect of agricultural aid on $sum_{forest\ NDVI}$ by types of agricultural aid.

| | **Model 1** | | | | | | |
|---|---|---|---|---|---|---|---|
| | **ILCSSSDRC** | **PAGE** | **RRA** | **RFCIP** | **SNAP** | **EWSPD** | **DFP** |
| **Fixed effects** | | | | | | | |
| D (Receiving aid) | −0.005 | −0.008 * | −0.008 | 0.493 | −0.178 | −0.001 | −0.612 |
| | (0.004) | (0.003) | (0.007) | (0.671) | (0.113) | (0.007) | (0.879) |
| T (Time) | −0.013 *** | 0.023 *** | 0.003 *** | 0.004 | 0.134 *** | 0.014 *** | 0.186 *** |
| | (0.001) | (0.001) | (0.001) | (0.012) | (0.004) | (0.001) | (0.016) |
| D × T (Receiving aid × Time) | 0.031 *** | −0.021 *** | −0.005 *** | −0.035 ** | −0.125 *** | −0.018 *** | −0.193 *** |
| | (0.001) | (0.001) | (0.001) | (0.016) | (0.006) | (0.001) | (0.023) |
| Constant | −0.453 | −0.456 *** | −0.456 *** | −0.074 | −0.203 * | −0.452 *** | 0.163 |
| | (0.003) | (0.002) | (0.005) | (0.555) | (0.113) | (0.004) | (0.622) |
| **Random effects** | | | | | | | |
| Chiefdom | $6.642 \times 10^{-6}$ | $5.900 \times 10^{-6}$ | $2.238 \times 10^{-5}$ | 2.150 | 0.051 | $4.664 \times 10^{-5}$ | 0.386 |
| | (0.003) | (0.002) | (0.005) | (1.467) | (0.225) | (0.006) | (0.621) |
| Time | $6.009 \times 10^{-6}$ | $6.467 \times 10^{-6}$ | $7.290 \times 10^{-7}$ | 0.038 | 0.003 | $4.515 \times 10^{-6}$ | 0.006 |
| | (0.002) | (0.003) | (0.001) | (0.196) | (0.053) | (0.002) | (0.078) |
| Residual | $1.482 \times 10^{-5}$ | $1.012 \times 10^{-4}$ | $3.653 \times 10^{-6}$ | 0.148 | 0.011 | $4.153 \times 10^{-5}$ | 0.021 |
| | (0.004) | (0.010) | (0.002) | (0.385) | (0.103) | (0.002) | (0.143) |
| **Number of Observation** | 460 | 920 | 92 | 8940 | 4140 | 792 | 648 |

Note: * $p < 0.1$; ** $p < 0.05$; *** $p < 0.01$. ILCSSSDRC: Improving Living Conditions and Strengthening of Social Stability and Decentralization in Rural Communities of Former Civil War Areas in Southern Sierra Leone; PAGE: Promoting Agriculture, Governance and the Environment; RRA: Refugees in Rural Areas in Sierra Leone; RFCIP: Rural Finance and Community Improvement Project; SNAP: Sustainable Nutrition and Agricultural Promotion; EWSPD: Empowering Women for Sustainable Peace and Development; DFP: Diversified Food Production.

## 4. Conclusions

How to make foreign aid more environmentally friendly and to avoid disrupting biodiversity in the recipient countries are critical issues, yet they have not been discussed widely enough in the global governance of aid. This study empirically explored the effects of agricultural aid on forests by conducting a sub-national analysis based on remotely sensed data of Sierra Leone. Firstly, the findings demonstrate that agricultural aid overall degrades forests in Sierra Leone. The results highlight the conflict between food production and nature conservation, and thus the need to balance development and environmental

preservation in the sub-Saharan region. This may be partially explained by the reality of poverty and the low quality of governance in Sierra Leone. For most of the population, rice harvested during harvesting period (from September to January) is highly insufficient to meet the needs of their family for the whole year. For instance, in 2015, only 4% of farmers are able to meet their rice needs for twelve months, while more than 66% of farmers can only meet their rice needs for six or less than six months (CFSV, 2015). Without additional non-farm jobs for earning a living and with a low quality of governance, agricultural aid may be mainly used or abused to conduct more commercially valuable activities, such as cutting trees for timber, firewood, and charcoal, which are major drivers of forest degradation and deforestation.

Secondly, the relationship between agricultural aid and forest growth does not change greatly over time. We do not find a shift in the relationship between agricultural aid and forest growth. Agricultural aid negatively and significantly influences forest growth both in the short term (i.e., 16-day frequency) and long term (i.e., years), and a turning point does not appear at least during the period from 2001 to 2015. All together, these results indicate that the negative impact of agricultural aid on forests will not diminish automatically over time. Continued interventions are needed to prevent agricultural aid from driving forest degradation.

The results also show that different kinds of agricultural aid have different impacts on forests. Agricultural aid without specific objectives (SNAP), and with agricultural growth and production as the major goal (DFP and RFCIP) damage forests the most. Unfortunately, these three types of aid make up the largest proportion of all aid to agricultural and rural development (Table 1). On the contrary, those few aid projects that take account of social and environmental concerns (ILCSSSDRC, PAGE and EWSPD) are found to damage forests less. The agricultural aid of ILCSSSDRC, which aims to improve the quality of governance, benefited the forests significantly. According to Lambin's work (2009) on forest transition, endogenous socio-ecological feedbacks seem to better explain a slowing down of the deforestation and stabilization of forest cover, and exogenous socio-economic factors better explain reforestation [59], while our findings demonstrate that agricultural aid as an exogenous factor accounts for both forest decline and recovery. This implies that the causal relationship between agricultural development and forests is not based on an intrinsic tension, so that we do not have to sacrifice one for the other. Rather, how the policies are made and how agricultural aid is implemented can make a real difference in forest outcomes.

These findings clearly have implications for policy opportunities in protecting forests in the sub-Saharan region when implementing agricultural aid. In 2018, the World Bank issued a specific report in which scholars and experts proposed that improving the quality of governance is one of the two foundational ways to accelerate growth, poverty reduction, and shared prosperity in Sierra Leone [50]. The implications from this study are in line with the World Bank's suggestion. Our findings show that among seven different kinds of agricultural aid, only the one focusing on social stability and decentralization in rural communities (ILCSSSDRC) positively influenced forest growth conditions. This implies that better governance is of key importance for forest protection in contemporary Sierra Leone. Therefore, increasing agricultural aid to rural governance may benefit forests in the long run. Additionally, although agricultural aid with environmental and other social concerns harms forests (i.e., PAGE and EWSPD), their impacts are smaller compared to that of agricultural aid with agricultural development and poverty reduction as major objectives or without specific objectives (i.e., DFP, RFCIP, and SNAP). As such, forests will benefit from increases in the proportion of social- and environment-oriented agricultural aid projects. Currently, these programs are not a priority.

Several areas of future work seem promising. Firstly, in this study, we assessed the impact of agricultural aid on forests by examining only one country as a case study. More analyses over a broader geographic region are required to better understand the relationships between agricultural aid and forests. Secondly, we differentiated different types of

agricultural aid based on the names and purposes of agricultural aid projects categorized by AidData. However, because an agricultural aid project may involve several objectives, merely relying on projects' name and purposes may not perfectly distinguish the types of agricultural aid. Further research is needed to differentiate distinct types of agricultural aid more reliably. Thirdly, this study only investigated the impact of exposure to agricultural aid on forest growth. However, the effect may vary by the amount of agricultural aid to different chiefdoms. More research is required to assess the environmental outcomes of agricultural aid. Lastly, the difference-in-differences design is a valuable tool in evaluating various development policies, but its key assumption of parallel trends, which justifies that the DID estimator is unbiased, is difficult to perfectly satisfy in many applications. This study tries to address this challenge by matching the geographically closest chiefdoms in the same district and controlling for chiefdom-specific and time-specific characteristics. With more high frequency data at the chiefdom level (e.g., precipitation, area in cultivation, level of agricultural development), studies can further improve the DID design for characterizing the relationship between agricultural aid and forest degradation.

**Author Contributions:** Conceptualization, Q.H.; methodology, Q.M., W.F., Y.H. and Q.H.; formal analysis, Q.M., Y.H. and Q.H.; data curation, Q.M., W.F. and Q.H.; writing—original draft preparation, Q.M. and Q.H.; writing—review and editing, W.F., Y.H. and Q.H.; visualization, Q.M., Y.H. and Q.H.; supervision, Q.H. All authors have read and agreed to the published version of the manuscript.

**Funding:** Q.H. was supported by the Major Program of the National Social Science Foundation of China (Grant Number 17ZDA110).

**Institutional Review Board Statement:** Not applicable.

**Informed Consent Statement:** Not applicable.

**Data Availability Statement:** Geo-coded agricultural aid data of Sierra Leone were downloaded from https://www.aiddata.org/datasets (accessed on 1 July 2021), Moderate-resolution Imaging Spectroradiometer (MODIS) land cover dataset (MCD12Q1) is available through https://lpdaac.usgs.gov/products/mcd12q1v006/ (accessed on 1 August 2021), and Normalized Difference Vegetation Index (NDVI) data from MODIS Vegetation Indices product (MOD13A1) were from https://modis.gsfc.nasa.gov/data/dataprod/mod13.php (1 August 2021).

**Acknowledgments:** Q.H. acknowledges financial support from the Major Program of the National Social Science Foundation of China (Grant Number 17ZDA110). The authors gratefully acknowledge Xun Pang from Tsinghua University, China, for providing thoughtful suggestions and comments. The authors also extend great gratitude to the anonymous reviewers and editors for their critical comments and help.

**Conflicts of Interest:** The authors declare no conflict of interest.

## Appendix A

**Table A1.** Causal effect of agricultural aid on forests with control variables of temperature and precipitation.

| | Model 1 | |
|---|---|---|
| | **Estimate** | **Std. Error** |
| **Fixed effects** | | |
| D (Receiving aid) | 0.139 | 0.294 |
| T (Time) | 0.040 *** | 0.005 |
| D × T (Receiving aid × Time) | −0.079 *** | 0.006 |
| Temperature | −0.011 *** | 0.001 |
| Precipitation | −0.001 *** | 0.000 |
| Intercept | −0.216 | 0.269 |

**Table A1.** *Cont.*

| | Model 1 | |
|---|---|---|
| | **Estimate** | **Std. Error** |
| **Random effects** | | |
| | **Variance** | **Std. Dev** |
| Chiefdom | 0.903 | 0.950 |
| Time | 0.003 | 0.058 |
| Residual | 0.076 | 0.275 |
| **Number of Observation** | 29,790 | |

Note: *** $p < 0.01$.

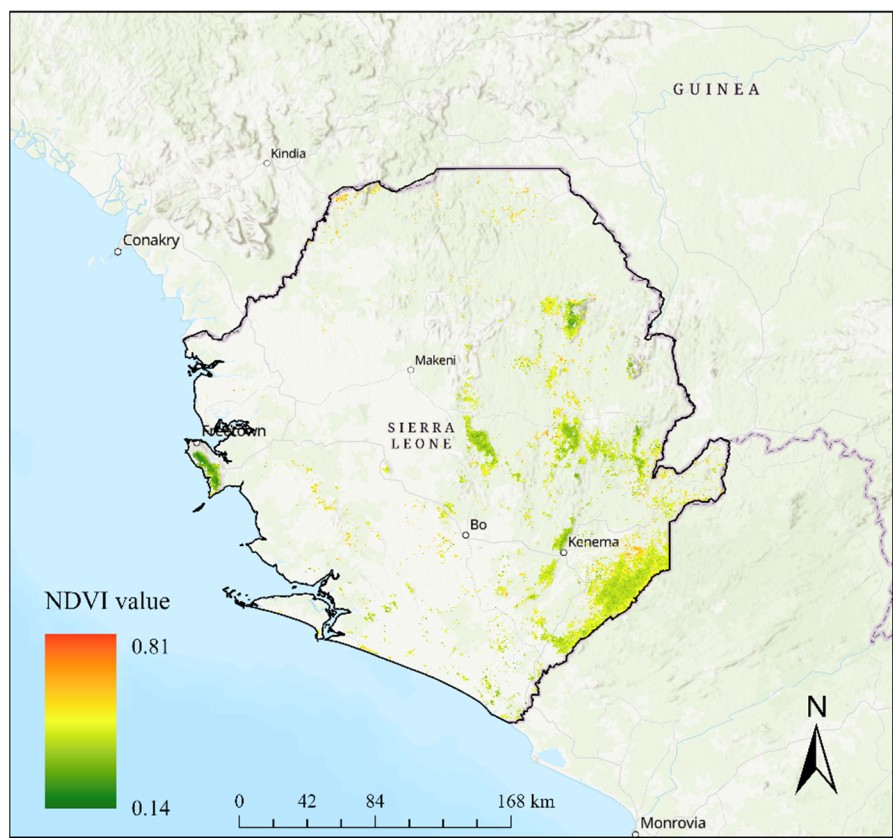

**Figure A1.** Mean NDVI values of forests during the growing season from June to August averaged from 2001 to 2015.

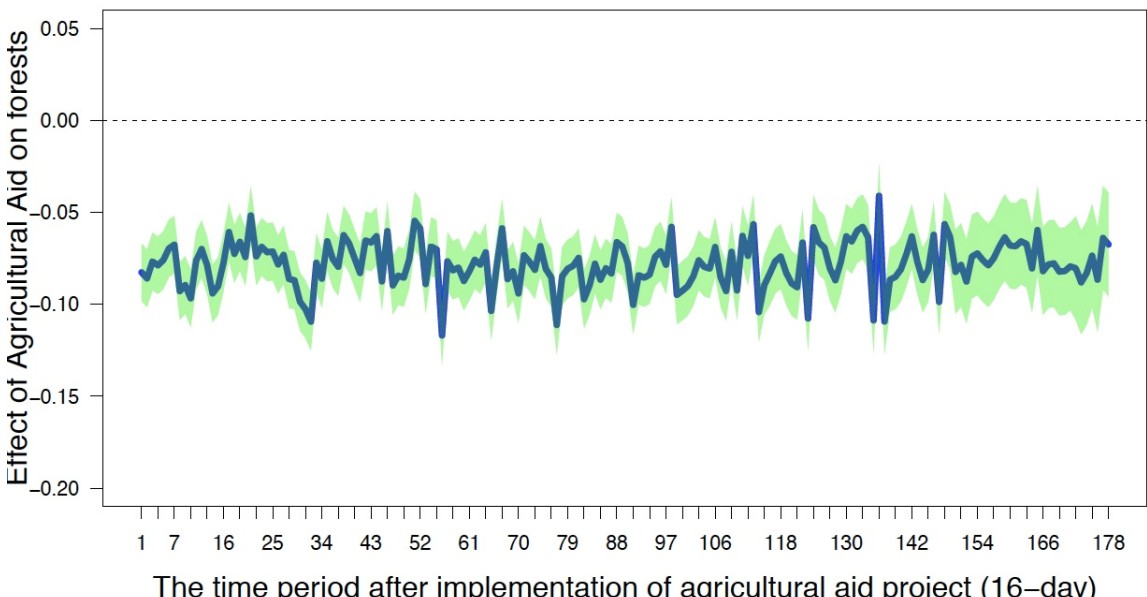

**Figure A2.** Causal effect of agricultural aid on $sum_{forest\ NDVI}$ across time with control variables of temperature and precipitation (the shaded green areas represent the 95% confidence interval).

**Table A2.** Causal effect of agricultural aid on $sum_{forest\ NDVI}$ by types of agricultural aid with the control variables of temperature and precipitation.

| | **Model 1** | | | | | | |
|---|---|---|---|---|---|---|---|
| | **ILCSSSDRC** | **PAGE** | **RRA** | **RFCIP** | **SNAP** | **EWSPD** | **DFP** |
| **Fixed effects** | | | | | | | |
| D (Receiving aid) | −0.005 | −0.008 * | −0.008 | 0.537 | −0.1771 | −0.001 | −0.608 |
| | (0.006) | (0.002) | (0.007) | (0.747) | (0.135) | (0.007) | (0.467) |
| T (Time) | −0.013 *** | 0.023 *** | 0.003 *** | 0.030 ** | 0.135 *** | 0.014 *** | 0.184 *** |
| | (0.001) | (0.001) | (0.001) | (0.013) | (0.005) | (0.001) | (0.016) |
| D × T (Receiving aid × Time) | 0.031 *** | −0.021 *** | −0.005 *** | −0.063 ** | −0.124 *** | −0.018 *** | −0.193 *** |
| | (0.001) | (0.001) | (0.001) | (0.018) | (0.006) | (0.001) | (0.023) |
| Temperature | 0.000 | 0.000 | 0.000 | −0.000 | −0.010 *** | 0.000 | 0.001 |
| | (0.000) | (0.000) | (0.000) | (0.000) | (0.003) | (0.000) | (0.001) |
| Precipitation | $-1.15 \times 10^{-5}$ *** | $-9.611 \times 10^{-6}$ *** | $-3.743 \times 10^{-6}$ *** | $-8.084 \times 10^{-4}$ *** | $-2.611 \times 10^{-4}$ *** | $-4.774 \times 10^{-6}$ *** | $-4.036 \times 10^{-4}$ *** |
| | (0.000) | (0.000) | (0.001) | (0.000) | (0.000) | (0.000) | (0.000) |
| Constant | −0.453 | −0.462 *** | −0.459 *** | 0.340 | −0.203 * | −0.460 *** | 0.040 |
| | (0.003) | (0.008) | (0.009) | (0.663) | (0.113) | (0.012) | (0.362) |
| **Random effects** | | | | | | | |
| Chiefdom | $1.600 \times 10^{-5}$ | $5.247 \times 10^{-6}$ | $2.517 \times 10^{-5}$ | 2.347 | 0.051 | $4.587 \times 10^{-5}$ | 0.109 |
| | (0.004) | (0.002) | (0.005) | (1.532) | (0.225) | (0.007) | (0.330) |
| Time | $1.766 \times 10^{-6}$ | $4.182 \times 10^{-6}$ | $1.450 \times 10^{-8}$ | 0.014 | 0.001 | $2.791 \times 10^{-6}$ | 0.000 |
| | (0.001) | (0.002) | (0.001) | (0.119) | (0.033) | (0.002) | (0.000) |
| Residual | $1.497 \times 10^{-5}$ | $1.010 \times 10^{-4}$ | $3.710 \times 10^{-6}$ | 0.162 | 0.011 | $4.145 \times 10^{-5}$ | 0.020 |
| | (0.004) | (0.010) | (0.002) | (0.403) | (0.103) | (0.006) | (0.141) |

**Table A2.** *Cont.*

| | Model 1 | | | | | | |
|---|---|---|---|---|---|---|---|
| | **ILCSSSDRC** | **PAGE** | **RRA** | **RFCIP** | **SNAP** | **EWSPD** | **DFP** |
| **Number of Observation.** | 460 | 920 | 92 | 8046 | 4140 | 792 | 648 |

Note: * $p < 0.1$; ** $p < 0.05$; *** $p < 0.01$. ILCSSSDRC: Improving Living Conditions and Strengthening of Social Stability and Decentralization in Rural Communities of Former Civil War Areas in Southern Sierra Leone; PAGE: Promoting Agriculture, Governance and the Environment; RRA: Refugees in Rural Areas in Sierra Leone; RFCIP: Rural Finance and Community Improvement Project; SNAP: Sustainable Nutrition and Agricultural Promotion; EWSPD: Empowering Women for Sustainable Peace and Development; DFP: Diversified Food Production.

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
