# Peer review of "Examining the Effects of Agricultural Aid on Forests in Sub-Saharan Africa: A Causal Analysis Based on Remotely Sensed Data of Sierra Leone"

_land, doi:10.3390/land11050668_

Round 1

Author Response

Thanks for your comments. We have modified our manuscript accordingly. Please see the attached responding letter.

Reviewer 2 Report

This paper addresses a very interesting question about the side effects of  development aid programmes on natural resources and the environment. Through the use of data of the number of agricultural aid projects implemented at subnational level and the evolution of NDVI measured by remote sensed techniques, the authors intend to draw casual relationships between agricultural development and forest degradation.

The paper faces, however, some drawbacks that lead me to reject its publication and the current from. Some of the points that author should  consider before resubmitting  a new version are related to:

  1. The research has a serious problem of attribution.Many of the assumptions made by the authors to apply the difference-in-difference approach: the parallel time trends; the consideration of the multiple project as a only one treatment make not possible the attribution to changes observed to a unique factor: the implementation of aid projects. Based on this data is not possible to establish a “casual” links as argued by the authors (line 347)
  2. Besides that, references of the difference in difference approach are needed so the readers can check robustness and appropriateness of this method.
  3. The introduction and conclusion can be enriched with some references to and discussion on the application of the forest transition model that have be used to explain the evolution of forest in several developing countries ( see for example Eric F. Lambin, Patrick Meyfroidt, 2010 Land use transitions: Socio-ecological feedback versus socio-economic change, Land Use Policy, 27, 108-118 https://doi.org/10.1016/j.landusepol.2009.09.003).
  4. More references are needed about the source of data on aid projects: the Aid data. Who /which institution run and manage this data base? Are the data available ? Which type of information and standard procedures are used?

Author Response

(The authors gave the same response as above.)

Reviewer 3 Report

The manuscript is well written and interesting. It proves well-known aspect that the aid to agricuture influences the rate of deforestation at specific area.

Author Response

Thanks for your support of our manuscript.

Round 2

Reviewer 2 Report

I would like to recognize the effort made by the authors to improve the pervious version of the paper by addressing the suggestions and concerns raised by the reviewers.

They have provided a detailed explanation about from the data of agricultural aid development projects were sourced. As regards the method used for testing the effects of aid projects on the forest growth, the authors provide references that support the use of this methodology in the field of social sciences

Author Response

Dear reviewer, we appreciate you for supporting our work!